# DBT: A Detection Booster Training method for improving the accuracy of classifiers

## Abstract

Deep learning models owe their success at large, to the availability of a large amount of annotated data. They try to extract features from the data that contain useful information needed to improve their performance on target applications. Most works focus on directly optimizing the target loss functions to improve the accuracy by allowing the model to implicitly learn representations from the data. There has not been much work on using background/noise data to estimate the statistics of in-domain data to improve the feature representation of deep neural networks. In this paper, we probe this direction by deriving a relationship between the estimation of unknown parameters of the probability density function (pdf) of input data and classification accuracy. Using this relationship, we show that having a better estimate of the unknown parameters using background and in-domain data provides better features which leads to better accuracy. Based on this result, we introduce a simple but effective detection booster training (DBT) method that applies a detection loss function on the early layers of a neural network to discriminate in-domain data points from noise/background data, to improve the classifier accuracy. The background/noise data comes from the same family of pdfs of input data but with different parameter sets (e.g., mean, variance). In addition, we also show that our proposed DBT method improves the accuracy even with limited labeled in-domain training samples as compared to normal training. We conduct experiments on face recognition, image classification, and speaker classification problems and show that our method achieves superior performance over strong baselines across various datasets and model architectures.

## 1 Introduction

Modern pattern recognition systems achieve outstanding accuracies on a vast domain of challenging computer vision, natural language, and speech recognition benchmarks (Russakovsky et al. (2015); Lin et al. (2014); Everingham et al. (2015); Panayotov et al. (2015)). The success of deep learning approaches relies on the availability of a large amount of annotated data and on extracting useful features from them for different applications. Learning rich feature representations from the available data is a challenging problem in deep learning. A related line of work includes learning deep latent space embedding through deep generative models (Kingma & Welling (2014); Goodfellow et al. (2014); Berthelot et al. (2019) or using self-supervised learning methods (Noroozi & Favaro (2016); Gidaris et al. (2018); Zhang et al. (2016b)) or through transfer learning approaches (Yosinski et al. (2014); Oquab et al. (2014); Razavian et al. (2014)).

In this paper, we propose to use a different approach to improve the feature representations of deep neural nets and eventually improve their accuracy by estimating the unknown parameters of the probability density function (pdf) of input data. Parameter estimation or Point estimation methods are well studied in the field of statistical inference (Lehmann & Casella (1998)). The insights from the theory of point estimation can help us to develop better deep model architectures for improving the model's performance. We make use of this theory to derive a correlation between the estimation of unknown parameters of pdf and classifier outputs. However, directly estimating the unknown pdf parameters for practical problems such as image classification is not feasible since it can sum up to millions of parameters. In order to overcome this bottleneck, we assume that the input data points are sampled from a family of pdfs instead of a single pdf and propose to use a detection based training approach to better estimate the unknowns using in-domain and background/noise data. One alternative is that we can use generative models for this task, however, they mimic the general

46 distribution of training data conditioned on random latent vectors and hence cannot be directly applied
47 for estimating the unknown parameters of a family of pdfs. Our proposed detection method involves
48 a binary class discriminator that separates the target data points from noise or background data. The
49 noise or background data is assumed to come from the same family of distribution of in-domain
50 data but with different moments (Please refer to the appendix for more details about the family of
51 distributions and its extension to a general structure). In image classification, this typically represents
52 the background patches from input data that fall under the same distribution family. In speech domain,
53 it can be random noise or the silence intervals in speech data. Collecting such background data to
54 improve the feature representations is much simpler as compared to using labeled training data since
55 it is time-consuming and expensive to collect labeled data. Since the background patches in images
56 or noise in speech signals are used for binary classification in our method, we refer to such data
57 as the noise of an auxiliary binary classification problem denoted by auxiliary binary classification
58 (ABC)-noise dataset. An advantage of using ABC-noise data during training is that it can implicitly
59 add robustness to deep neural networks against the background or noisy data.

60 Since ABC-noise data can be collected in large quantities for free and using that data in our approach
61 improves the classification benchmarks, we investigate whether this data can act as a substitute for
62 labeled data. We conduct empirical analysis and show that using only a fraction of labeled training
63 data together with ABC-noise data in our DBT method, indeed improves the accuracy as compared
64 to normal training.

65 To summarize, our contributions are threefold. First, we present a detailed theoretical analysis on
66 the relation between the estimation of unknown parameters of pdf of data and classification outputs.
67 Second, based on the theoretical analysis, we present a simple booster training method to improve
68 classification accuracy which also doubles up as an augmented training method when only limited
69 labeled data is available. Third, we consistently achieve improved performances over strong baselines
70 on face recognition, image classification, and speaker recognition problems using our proposed
71 method, showing its generalization across different domains and model architectures.

## 2 RELATED WORK

73 **Notations and Preliminary:** In this paper, vectors, matrices, functions, and sets are denoted by bold
74 lower case, bold uppercase, lower case, and calligraphic characters, respectively. Consider a datapoint
75 denoted by $\mathbf{x}$. We assume that $\mathbf{x}$ belongs to a family of probability density functions (pdf's) defined
76 as $\mathcal{P} = \{p(\mathbf{x}, \boldsymbol{\theta}), \boldsymbol{\theta} \in \Theta\}$, where $\Theta$ is the possible set of parameters of the pdf. In general, $\boldsymbol{\theta}$ is a real
77 vector in higher dimensions. For example, in a mixture of Gaussians, $\boldsymbol{\theta}$ is a vector containing the
78 component weights, the component means, and the component covariance matrices. In this paper, we
79 assume that $\boldsymbol{\theta}$ is an unknown deterministic function (There are other approaches such as bayesian
80 that consider $\boldsymbol{\theta}$ as a random vector). In general, although the structure of the family of pdfs is itself
81 unknown, defining a family of pdfs such as $\mathcal{P}$ can help us to develop theorems and use those results
82 to derive a new method. For the family of distribution $\mathcal{P}$, we can define the following classification
83 problem

$$\{ \ \mathcal{C}_1 : \boldsymbol{\theta} \in \Theta_1, \mathcal{C}_2 : \boldsymbol{\theta} \in \Theta_2, \cdots, \mathcal{C}_n : \boldsymbol{\theta} \in \Theta_n \ \} \tag{1}$$

84 where set of $\Theta_i$'s is a partition of $\Theta$. The notation of (1) means that, class $\mathcal{C}_i$ deals with a set of
85 data points whose pdf is $p(\mathbf{x}, \boldsymbol{\theta}_i)$ where $\boldsymbol{\theta}_i \in \Theta_i$. A wide range of classification problems can be
86 defined using (1) e.g., ((Lehmann & Casella, 2006, Chapter 3)) and ((Duda et al., 2012, Chapter 4)).
87 The problem of estimating $\boldsymbol{\theta}$ comes under the category of parametric estimation or point estimation
88 (Lehmann & Casella (1998)). Estimating the unknown parameters of a given pdf $p(\mathbf{x}, \boldsymbol{\theta})$, have been
89 extensively studied in the field of point estimation methods (Lindgren (2017); Lee et al. (2018);
90 Lehmann & Casella (2006)). An important estimator in this field is the minimum variance unbiased
91 estimator and it is governed by the Cramer Rao bound. The Cramer Rao bound provides the lower
92 bound of the variance of an unbiased estimator (Bobrovsky et al. (1987)). Let the estimation of
93 $\boldsymbol{\theta}$ be denoted by $\widehat{\boldsymbol{\theta}}$, and assume that $\widehat{\boldsymbol{\theta}}$ is an unbiased estimator, i.e., $E(\widehat{\boldsymbol{\theta}}) = \boldsymbol{\theta}$. Its covariance
94 matrix denoted by $\boldsymbol{\Sigma}_{\widehat{\boldsymbol{\theta}}}$ satisfies $\boldsymbol{\Sigma}_{\widehat{\boldsymbol{\theta}}} - \mathbf{I}^{-1}(\boldsymbol{\theta}) \succeq \mathbf{0}$, where $\mathbf{A} \succeq \mathbf{0}$ implies that $\mathbf{A}$ is a non-negative
95 definite matrix ((Lehmann & Casella, 1998, chapter 5)) and $\mathbf{I}(\boldsymbol{\theta}) := -E(\partial^2 \log(p(\mathbf{x}, \boldsymbol{\theta}))/\partial \boldsymbol{\theta}^2)$
96 is called the Fisher information matrix. For an arbitrary differentiable function $g(\cdot)$, an efficient
97 estimator of $\mathbf{g}(\boldsymbol{\theta})$ is an unbiased estimator when its covariance matrix equals to $\mathbf{I}_{\mathbf{g}}^{-1}(\boldsymbol{\theta})$, where $\mathbf{I}_{\mathbf{g}}^{-1}(\boldsymbol{\theta})$
98 is the fisher information matrix of $\mathbf{g}(\boldsymbol{\theta})$, i.e., the efficient estimator achieves the lowest possible

variance among all unbiased estimators. The efficient estimator can be achieved using factorization of $\partial \log(p(\mathbf{x}, \boldsymbol{\theta}))/\partial \mathbf{g}(\boldsymbol{\theta}) = I_{\mathbf{g}}(\boldsymbol{\theta})(\widehat{\mathbf{g}}(\mathbf{x}) - \mathbf{g}(\boldsymbol{\theta}))$, if it exists (Rao (1992); Lehmann & Casella (1998)). Based on these results, we derive a relationship between the efficient estimation of unknowns and maximum likelihood classifier of (1) and use auxiliary binary classifiers to apply that result in our proposed DBT method.

**Parameter Estimations:** Independent component analysis (Hyvärinen (1999)) decomposes a multivariate signal into independent non-Gaussian signals. ICA can extract non-Gaussian features from Gaussian noise. Additionally, there is a class of classifiers called generalized likelihood ratio functions that replaces the estimation of unknown parameters into the likelihood functions. This approach provides a huge improvement in the field of parametric classifiers, where the family of pdf of data is given (Zeitouni et al. (1992), Conte et al. (2001), Lehmann & Casella (2006)). Noise-contrastive estimation (NCE) (Gutmann & Hyvärinen (2010)) involves training a generative model that allows a model to discriminate data from a fixed noise distribution. Then, this trained model can be used for training a sequence of models of increasing quality. This can be seen as an informal competition mechanism similar in spirit to the formal competition used in the adversarial networks game. In Bachman et al. (2019), a feature selection is proposed by maximizing the mutual information of the difference between features extracted from multiple views of a shared context. In that work, it is shown that the best results is given by using a mutual information bound based on NCE. The key difference between our method and NCE is that, we do not construct a generative model for noise. Instead of estimating the pdf of noise in NCE, we estimate the parameters of pdf of in-domain dataset using an auxiliary class that has many common parameters in its pdf. Moreover, we show that the estimation of that parameters are sufficient statistic for a classifier. We assume that the noise dataset is not pure and it has some similarity with the in-domain dataset, where it can help the feature selection layers to select relevant (in-domain) features, e.g., see Fig. 3. Further, in our approach, we do not construct the pdf of noise or in-domain data, instead we estimate its parameters directly, which is more efficient in terms of training, computation and also dimensionality reduction.

Auxiliary classifiers were introduced in inception networks (Szegedy et al. (2015)) and used in (Lee et al. (2015); S. et al. (2016)) for training very deep networks to prevent vanishing gradient problems. Further, auxiliary classifiers were also proposed for early exit schemes (Teerapittayanon et al. (2016)) and self-distillation methods (Zhang et al. (2019a;b)). Such auxiliary classifiers tackle different problems by predicting the same target as the final classification layer. In contrast, our proposed DBT method involves auxiliary binary classifiers that detect noise, interference, and/or background data from in-domain data points for improving the target classification accuracy.

## 3 ESTIMATION OF PARAMETERS OF PDF AND CLASSIFICATION

For (1), we define a deterministic discriminative function of $\Theta_i$, denoted by $t_i(\cdot)$ such that the following conditions are satisfied:

- $t_i(\cdot)$ maps $\Theta$ to real numbers such that $t_i(\boldsymbol{\theta}) > 0$, if $\boldsymbol{\theta} \in \Theta_i$ and $t_i(\boldsymbol{\theta}) \leq 0$ for $\boldsymbol{\theta} \notin \Theta_i$.
- $t_i(\cdot)$ is a differentiable function almost everywhere and $\int_{\Theta} |t_i(\boldsymbol{\theta})| \mathrm{d}\mu_l(\boldsymbol{\theta}) < \infty$, where $\mu_l$ denotes the Lebesgue measure.

The following theorem shows the relationship of $t_i(\cdot)$ and the log-likelihood ratio of class $\mathcal{C}_i$ versus other classes. The proofs of Theorems 1, 2 and 3 are provided in the appendix.

**Theorem 1** *Assume that the pdf $p(\mathbf{x}, \boldsymbol{\theta})$ is differentiable with respect to $\boldsymbol{\theta}$ almost everywhere. If the efficient minimum variance and unbiased estimation of a deterministic discriminative function of $\Theta_i$ exists, then the log likelihood ratio of class $i$ against the rest of classes is an increasing function of the minimum variance and unbiased estimation of $\Theta_i$.*

Directly from this theorem, it follows that the optimal classifier using the maximum likelihood for (1) is given as follows $d(\mathbf{x}) = \arg\max_{i \in \{1, \cdots, n\}} k_i(\widehat{t_i}(\mathbf{x}))$, where $k_i$'s are some increasing functions and $t_i(\cdot)$'s are the deterministic discriminative function of $\Theta_i$'s such that the efficient minimum variance and unbiased estimation for them exists. Based on this result, a set of minimum variance and unbiased estimation of deterministic discriminative functions of $\Theta_i$'s leads us to the maximum likelihood classifier. One approach is to directly estimate the deterministic discriminative functions, instead of maximizing the likelihood function. However, finding deterministic discriminative functions that have efficient minimum variance and unbiased estimation may not be feasible in practical problems,

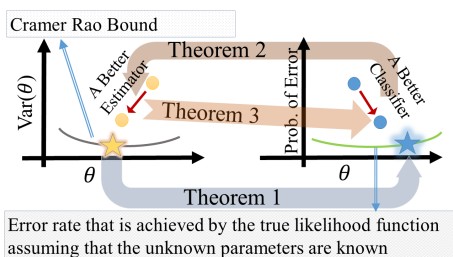

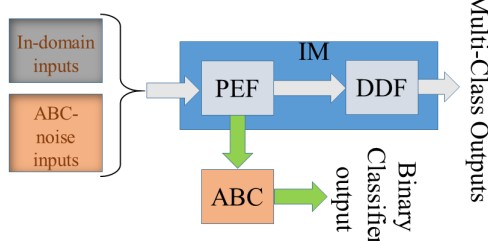

Figure 1: Visualizing Theorems 1,2 and 3

Figure 2: A general schema of our proposed DBT method with PEF, DDF and ABC blocks

especially when the dimension of $\boldsymbol{\theta}$ increases. Theorems 2 and 3 study the same relationship between the estimation of unknown parameters and the accuracy of classifiers for sub-optimal estimators and classifiers.

**Theorem 2** *Consider the output of two classifiers for the ith class as follows: $r_j(\mathbf{x}) = i$ if $h_j(\mathbf{x}) > \tau$ and $r_j(\mathbf{x}) = $ other classes if $h_j(\mathbf{x}) < \tau$, where $j \in \{1, 2\}$. where $h_j(\mathbf{x})$ is the estimation of a deterministic discriminative function and $\tau$ is a classification threshold. Assume that the cumulative distribution function of $h_j(\mathbf{x})$'s have bounded inflection points, and also, the probability of true positive of $r_j(\mathbf{x})$ is an increasing function of $d(\boldsymbol{\theta})$, which is the deterministic discriminative function of class $i$, for all $i$. Further assume that for each $\tau$ the probability of false positive of $r_1(\mathbf{x})$ is less than the probability of false positive of $r_2(\mathbf{x})$ and the probability of true positive of $r_1(\mathbf{x})$ is greater than the probability of true positive of $r_2(\mathbf{x})$. Then, there exists a $h_{\min}$ such that for all $d(\boldsymbol{\theta}) > h_{\min}$ and all $\boldsymbol{\theta}$ we have $\Pr(|h_1(\mathbf{x}) - d(\boldsymbol{\theta})| < \epsilon) > \Pr(|h_2(\mathbf{x}) - d(\boldsymbol{\theta})| < \epsilon)$.*

Theorem 2 shows that a better classifier leads to a better estimation of $d(\boldsymbol{\theta})$. In the next theorem, we show the dual property of this result.

**Theorem 3** *Let $\Theta_m$ be a Borel set with positive Lebesgue measure in (1) for all $m \in \{1, \cdots, n\}$. Assume that $r_1(\cdot)$ and $r_2(\cdot)$ are given as follows $r_1(\mathbf{x}) = m$, if $\widehat{\boldsymbol{\theta}}_1 \in \Theta_m$ and $r_2(\mathbf{x}) = m$, if $\widehat{\boldsymbol{\theta}}_2 \in \Theta_m$. Also, assume that $\Pr(\|\widehat{\boldsymbol{\theta}}_1 - \boldsymbol{\theta}\| \leq \epsilon) \geq \Pr(\|\widehat{\boldsymbol{\theta}}_2 - \boldsymbol{\theta}\| \leq \epsilon)$, for all $\boldsymbol{\theta} \in \Theta = \cup_{m=1}^{n}\Theta_m$ and $\epsilon > 0$, then the probability of classification error $r_1(\cdot)$ is less than $r_2(\cdot)$ where $\widehat{\boldsymbol{\theta}}_1$ and $\widehat{\boldsymbol{\theta}}_2$ are two different estimators of $\boldsymbol{\theta} \in \Theta = \cup_{m=0}^{M-1}\Theta_m$.*

Theorem 3 proves that a more accurate estimator leads to a classifier that has a lower probability of classification error. From Theorem 1, we can infer that a sufficient statistic for developing the maximum likelihood classification is $\widehat{t}_i(\mathbf{x})$, which is the efficient minimum variance and unbiased estimation of the deterministic discriminative functions of $\Theta_i$'s denoted by $t_i(\boldsymbol{\theta})$. In other words, the maximum likelihood classifier is a function of $\mathbf{x}$ only via the efficient minimum variance and unbiased estimation $t_i(\boldsymbol{\theta})$. We can estimate $t_i(\boldsymbol{\theta})$ by replacing the estimation $\boldsymbol{\theta}$ in $t_i(\cdot)$, i.e., $\widehat{t_i(\boldsymbol{\theta})} \approx t_i(\widehat{\boldsymbol{\theta}})$, where $\widehat{\boldsymbol{\theta}}$ is a function of $\mathbf{x}$. From the above theorems, we conclude that improving the estimation of unknown parameters of pdf of data can improve the accuracy of the classifier. On the other side, having a good classifier means having a good estimator of unknowns of the pdf of input data. In many practical problems, the optimal maximum likelihood classifier may not be achievable, but the likelihood function of the classifier provides an optimal bound of the probability of error. In such cases, we can improve the accuracy of sub-optimal classifiers and that is the main focus of this paper. Fig. 1 illustrates the proposed theorems visually.

## 4 PROPOSED METHOD: DETECTION BOOSTER TRAINING (DBT)

In this section, we propose the *detection booster training (DBT)* method based on the achieved theorems in the previous section to improve the accuracy of deep networks. Specifically, we divide a deep model into two parts - early and later layers. We apply a detector (detection here means detecting a target pattern from noise/background) on the early layers of the neural network in order

| Loss | Ver. Acc. (%) |
|------|---------------|
| ResNet-50-DBT (CE) | 98.96 |
| ResNet-50-DBT | **99.12** |

Table 1: Verification accuracy on LFW dataset for two different $\mathcal{L}_{\mathrm{ABC}}$ trained using CASIA Yi et al. (2014) dataset.

| Loss | Acc. | Acc. on H-set |
|------|------|---------------|
| ResNet-100-AF | 78.85 | 00.04 |
| ResNet-100-DBT | **81.11** | **21.00** |

Table 2: Comparison of Rank-1 identification accuracy on the IJB-B, with animal distractors.

to improve the estimation of unknown parameters of the family of pdf (based on Theorem 2). A better estimation of unknown parameters corresponds to better feature representations in the early layers and these features are input to the rest of the layers to construct the deterministic discriminative functions (DDF) useful for the in-domain data classification (based on Theorem 3).

A general schema for dividing a deep model into two sub-models namely PEF (parameter estimator functions) and DDF is depicted in Figure 2. The early layers of the model estimate the unknown parameters of pdf of data while the later layers construct the discriminative functions essential for classification. Based on this scheme, we formally define the three main components of DBT as follows:

- *parameter estimator functions* (PEF): The sub-network from input layer to the $k$th layer, where $k$ is a hyperparameter in the DBT approach.
- *auxiliary binary classification* (ABC): Some additional layers are attached to the end of PEF, mapping the output of the $k$th layer to a one-dimensional vector.
- *deterministic discriminative functions* (DDF): The sub-network from $k$th layer to the output of the model. The output of model is a vector equal to the length of the number of classes $n$.

From Theorem 2, we showed that unknown parameter estimation can be improved using a detection approach. During training, we apply a binary classification on the early layers (PEF) of the model to improve the estimation of unknown parameters of pdf and subsequently provide rich feature vectors for DDF. We define the *auxiliary binary classification problem* (ABC problem) as follows:

- Class 1 (alternative hypothesis) of ABC problem denoted by $\mathcal{H}_1$ is set of all data points of classes of $\mathcal{C}_1$ to $\mathcal{C}_n$, i.e. $\boldsymbol{\theta} \in \cup_{i=1}^{n} \Theta_i$.
- Class 0 (null hypothesis) of ABC problem denoted by $\mathcal{H}_0$ is a dataset of data points from same distribution $p(\mathbf{x}, \boldsymbol{\theta})$ but $\boldsymbol{\theta} \notin \cup_{i=1}^{n} \Theta_i$. We also define the dataset of Class 0 of ABC as ABC-*noise* dataset, i.e., the ABC is given by the following hypothesis testing problem: $\mathcal{H}_1 : \boldsymbol{\theta} \in \cup_{i=1}^{n} \Theta_i$ versus $\mathcal{H}_0 : \boldsymbol{\theta} \notin \cup_{i=1}^{n} \Theta_i$. In many practical problems, the noise, background or interference data related to the in-domain dataset have same type of probability distribution but different pdf parameters. Hence, using that dataset is a cheap and adept choice for the null hypothesis of ABC.

The Auxiliary Binary Classification problem influences only the PEF and ABC units while the main classification problem with $n$ classes updates the parameters of both PEF and DDF using in-domain data. Since the auxiliary classifier is only used during training, the *inference model* (IM) consists of only PEF and DDF and hence, there is no additional computation cost during inference. We formulate the aforementioned method in the following notations and loss functions. Assume that $\mathbf{x}$ is a data point that belongs to Class $\mathcal{C}_i$, $i \in \{1, \cdots, n\}$ or Class $\mathcal{H}_0$ of ABC. Here, we define two type of labels denoted by $l_{\mathrm{ABC}}$ and $l_{\mathrm{MC}}$, where the subscription "MC" stands for multi-classes. So, if $\mathbf{x}$ belongs to class $\mathcal{C}_i$, then $l_{\mathrm{ABC}} = 1$ and $l_{\mathrm{MC}} = i - 1$, else if $\mathbf{x}$ is a ABC-noise data point, $l_{\mathrm{ABC}} = 0$ and $l_{\mathrm{MC}}$ is None. Therefore, the loss function is defined as:

$$\mathcal{L}_{\mathrm{tot}} = \mathcal{L}_{\mathrm{ABC}}(Q_{\mathrm{ABC}}(Q_{\mathrm{PEF}}(\mathbf{x})), l_{\mathrm{ABC}}) + \lambda l_{\mathrm{ABC}} \mathcal{L}_{\mathrm{MC}}(Q_{\mathrm{DDF}}(Q_{\mathrm{PEF}}(\mathbf{x})), l_{\mathrm{MC}}), \quad (2)$$

where $Q_{\mathrm{PEF}}, Q_{\mathrm{ABC}}$ and $Q_{\mathrm{DDF}}$ are the functions of PEF, ABC and DDF blocks, respectively. We set the hyperparameter $\lambda = 1$ to balance the two loss terms. It is seen that, the second term of the total loss is zero if $l_{\mathrm{ABC}} = 0$. $\mathcal{L}_{\mathrm{ABC}}$ and $\mathcal{L}_{\mathrm{MC}}$ are selected based on the problem definition and datasets. For classification, a simple selection for them can be binary cross-entropy and cross-entropy, respectively. For a given task and deep neural network, the choice of $k$ and $\mathcal{L}_{\mathrm{ABC}}$ influences the feature representation of early layers differently and consequently the accuracy of the model. We provide empirical studies in the next section to verify the same.

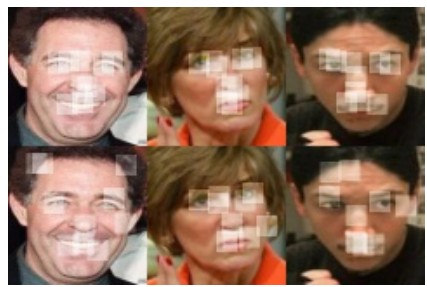

Figure 3: Maximally activated receptive fields of layer 15 of Inception-ResNet-v1 with (top row) and without (bottom row) DBT.

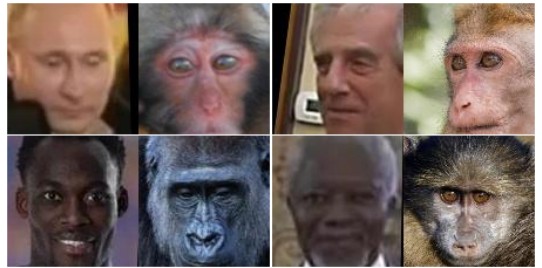

Figure 4: Examples of mis-identified faces along with their corresponding animal distractors on the IJB-B for ArcFace.

## 5 EXPERIMENTAL STUDY OF DBT

FACE RECOGNITION

We conduct experiments on face recognition benchmarks and show that the DBT method learns rich features essential for face recognition. We also discover an important observation that current state-of-the-art (SOTA) face recognition models are very sensitive to non-face data, in particular, animal faces. Fig. 4 shows a few examples of misidentified faces and their corresponding animal distractors from the IJB-B dataset using the ArcFace (Deng et al. (2019)) model. We show that our DBT method not only improves the verification accuracy but also implicitly tackles this robustness issue of current models against non-face data. Implementation details are provided in the appendix.

We consider the PEF discussed in Section 4 to be the first three layers of the model and DDF to be the rest of layers. Ablation studies on the choice of PEF and DDF are provided in the supplementary material. We define $\mathcal{L}_{\mathrm{MC}}$ in (2) as the SOTA ArcFace loss function proposed in (Deng et al. (2019)). The ABC-noise is a non-face dataset containing 500K images that we collected from background patches of MS1MV2 (Guo et al. (2016)) (More details in Appendix). We experimented with two different loss functions for $\mathcal{L}_{\mathrm{ABC}}$. For the first one, since popular face recognition models (Deng et al. (2019); Wang et al. (2018)) use normalized output features and compute the losses on a hypersphere, we select $\mathcal{L}_{\mathrm{ABC}}$ as follows. Let $p_f \in \mathbb{R}^d$ and $p_{nf} \in \mathbb{R}^d$ denote the prototypes for faces and non-faces, respectively. Following (Mettes et al. (2019)), we constrain the face/non-face prototypes on diametrically opposite directions i.e $\cos(\theta_{p_f p_{nf}}) = -1$ and normalize the output feature vectors for faces and non-faces such that $\|p_{f_i}\| = \|p_{nf_i}\| = 1$. We then define the $\mathcal{L}_{ABC}$ as,

$$\mathcal{L}_{\mathrm{ABC}} = -\frac{1}{N}\sum_{i=1}^{N}\log\big(\frac{e^{s(\cos(m_1\theta_{y_i}+m_2)-m_3)}}{e^{s(\cos(m_1\theta_{y_i}+m_2)-m_3)}+e^{s\cos\theta_2}}\big) + \frac{1}{N}\sum_{i=1}^{N}(-1-|p_{f_i}.p_{nf_i}|)^2, \qquad (3)$$

where $\theta_{y_i}$ and $\theta_2$ correspond to the angles between the weights and the features for face and non-face labels, respectively; $m_1, m_2, m_3$ are the angular margins; $s$ denotes the radius of the hypersphere. For the second choice, we use simple binary cross entropy for $\mathcal{L}_{\mathrm{ABC}}$. Table 1 shows that the verification accuracy on LFW (Huang et al. (2007)) using (3) is 0.16% higher than simple cross entropy loss. This also shows that choosing a task-specific $\mathcal{L}_{\mathrm{ABC}}$ is essential in obtaining more accurate results. We use Eqn.1 as the default for $\mathcal{L}_{\mathrm{ABC}}$ in all our face recognition experiments, unless otherwise stated.

Table 3 compares the verification accuracy of our method versus the current SOTA method ArcFace on five different test sets, LFW, CPLFW (Zheng & Deng (2018)), CALFW (Zheng et al. (2017)), CFP-FP (Sengupta et al. (2016)) and AgeDb-30 (Moschoglou et al. (2017)). For the LFW test set, we follow the unrestricted with labeled outside data protocol to report the performance. We trained ResNet-50 and ResNet-100 using ArcFace and DBT approaches on CASIA (small) and MS1MV2 (large) datasets, respectively. The results show that DBT method outperforms ArcFace on all datasets. Table 7 shows the angle statistics of the trained ArcFace and DBT models on the LFW dataset. Min. Inter and Inter refer to the mean of minimum angles and mean of all angles between the template embedding features of different classes (mean of the embedding features of all images for each class), respectively. Intra refers to the mean of angles between $x_i$ and template embedding feature for each class. From Table 7, we infer that DBT extracts better face features and hence reduces the intra-class variations. Directly from Tables 3 and 7, we infer that first, DBT consistently improves the accuracy

| Method | LFW | CALFW | CPLFW | CFPFP | AgeDb-30 |
|---|---|---|---|---|---|
| ResNet-50-AF (ArcFace) | 98.46 | 89.48 | 80.88 | 86.74 | 88.98 |
| ResNet-50-DBT | **99.12** | **91.38** | **87.10** | **94.95** | **91.23** |
| ResNet-100-AF (ArcFace) | 99.61 | 94.50 | 89.35 | 96.14 | 95.33 |
| ResNet-100-DBT | **99.75** | **95.13** | **90.70** | **96.90** | **96.16** |

Table 3: ArcFace vs. DBT-ArcFace: verification(%) accuracy on LFW, CALFW, CPLFW, CFP-FP and AgeDb-30 of models ResNet-100 and ResNet-50.

on all test sets. Second, learning better features in the early layers is crucial to obtain rich face feature embeddings. Third, the achieved gain using DBT is more pronounced on models trained using a smaller (CASIA) dataset (it has fewer identities and images). This shows that DBT can address the issue of the lack of in-domain data using cheap ABC-noise data.

We also provide the results of training Inception-ResNet-V1 and ResNet-64 models using DBT on MS1MV2 to show the generalization capacity of the DBT method. For the Inception-ResNet-V1 and ResNet-64, the PEF is set to be the first six layers and the DDF is the rest of the model. We use large margin cosine loss (LMCL) Wang et al. (2018) for $\mathcal{L}_{MC}$ and Cross entropy (CE) for $\mathcal{L}_{ABC}$. Table 4 shows the verification accuracy on LFW for Inception-ResNet-V1 and ResNet-64 models trained on MS1MV2 with and without DBT. The results show that DBT method is independent of model depth or architectures or loss functions and thereby consistently improves the accuracy compared to baseline results. Table 4 also compares the DBT method with state-of-the-art methods on LFW and YTF datasets. DBT method notably improves the baselines that are comparable to ArcFace and superior to all the other methods. We were not able to reproduce the results of the ArcFace paper using our Tensorflow implementation and dataset. We believe that using the original implementation and dataset from ArcFace will achieve superior results over the baselines on the benchmark datasets as evident from the results of our implementation. Finally, we compare the result ArcFace and DBT on IJB-B and IJB-C, in Table 5. It is seen that DBT provides a notable boost on both IJB-B and IJB-C by a considerable margin. DBT improves the verification accuracy as high as **1.94** % on IJB-B and **2.57** % on IJB-C dataset at $10^{-4}$ false alarm rate (FAR). We plot the receptive fields of the top ten maximally activated neurons of an intermediate layer of the face recognition model to visualize the features learned using the DBT method. Fig. 3 shows that the receptive fields of layer 15 of the inception-resnet-v1 model trained using DBT attends to the regions of eyes, nose and mouth as compared to insignificant regions in the normal training method. This shows that DBT learns more discriminative features essential to face recognition, corroborating our theoretical claims.

To show that current SOTA models are not robust to animal faces, we performed a 1:N identification experiment with approximately 3000 animal distractors on the IJB-B (Whitelam et al. (2017)) dataset. We trained the face recognition model with about 500K non-face data which contains 200 animal faces. This is disjoint from the 3000 distractors used in the identification experiment. We collected the animal faces from web images using MTCNN (Zhang et al. (2016a)) face detector which are the false positives from the face detector. Table 2 shows the Rank-1 identification accuracy of ResNet-100 on IJB-B dataset, trained on MS1MV2 using the ArcFace loss (ResNet-100-AF) versus our DBT approach (ResNet-100-DBT). The third column of Table 2 denotes the accuracy on a hard subset of images (false positives from ArcFace model) on the IJB-B dataset denoted by H-set. Results of Table 2 show that current face recognition models are unable to discriminate out-of-distribution (non-face) images from face images. Our ResNet-100-DBT significantly (as high as **21**%) reduces the misidentification rate as compared to the ArcFace model which shows that DBT method inherently overcomes this issue while also improving face recognition accuracy.

IMAGE CLASSIFICATION

In this section, we evaluate ResNet-110 and ResNext-101 models trained with and without DBT on image classification problem using CIFAR-10, CIFAR-100, and ImageNet. We also show the power of DBT to compensate for the smaller in-domain training set. For all implementations, PEF is defined to be the first three layers and DDF is the rest of the model. $\mathcal{L}_{ABC}$ and $\mathcal{L}_{MC}$ are set to cross-entropy loss. ABC-noise is the same data used in face recognition experiments. We follow the same training configurations from (He et al. (2016); Xie et al. (2017)).

To study the efficacy of the DBT method in augmenting smaller in-domain training datasets, we also trained ResNet-100 and ResNext-101 using partial training data on CIFAR-10 and CIFAR-100.

| Model | Loss | LFW | Method | LFW | YTF |
|---|---|---|---|---|---|
| Inception Resnet | CE | 99.45 | Center Loss | 99.28 | 94.9 |
| Inception Resnet-DBT | CE | **99.50** | Range Loss | 99.52 | 93.7 |
| Inception Resnet | LMCL | 99.55 | SphereFace | 99.42 | 95.0 |
| Inception Resnet-DBT | LMCL | **99.60** | SphereFace+ | 99.47 | - |
| Resnet 64 | CE | 99.55 | CosFace | 99.73 | 97.6 |
| Resnet 64-DBT | CE | **99.63** | ArcFace | **99.82** | **98.02** |
| Resnet 64 | LMCL | 99.65 | ArcFace** | 99.61 | 97.31 |
| Resnet 64-DBT | LMCL | **99.68** | ResNet-100-DBT | **99.75** | **97.67** |

Table 4: Comparison of DBT models with SOTA methods on LFW and YTF. ArcFace ** refers to our arcface implementation.

| Method | IJB-B | | | | | | IJB-C | | | | | |
|---|---|---|---|---|---|---|---|---|---|---|---|---|
| | $10^{-6}$ | $10^{-5}$ | $10^{-4}$ | $10^{-3}$ | $10^{-2}$ | $10^{-1}$ | $10^{-6}$ | $10^{-5}$ | $10^{-4}$ | $10^{-3}$ | $10^{-2}$ | $10^{-1}$ |
| ArcFace | 38.47 | 65.60 | 82.97 | 91.11 | 96.01 | 98.91 | 61.96 | 73.22 | 83.84 | 91.85 | 96.51 | **99.08** |
| DBT | **47.01** | **72.70** | **84.91** | **91.92** | **96.37** | **99.03** | **67.42** | **77.33** | **86.41** | **92.75** | **96.66** | 99.06 |

Table 5: 1:1 verification: ResNet-100: DBT vs. ArcFace on the IJB-B and IJB-C datasets

| Method | Top-1 | Top-5 |
|---|---|---|
| ResNet | 22.10 | 6.15 |
| ResNet-DBT | **21.82** | **6.02** |

Table 6: Top-1 and Top-5 error rates (%) on ILSVRC15 benchmark for ResNet w/o DBT.

| Method | Min. Inter | Intra | Inter |
|---|---|---|---|
| ArcFace | 53.23 | 7.2 | **88.73** |
| ResNet-DBT | **52.96** | **7.16** | 88.52 |

Table 7: Comparison of inter and intra angles (degrees) for different methods on LFW.

| ResNet Models | CIFAR-10 | CIFAR-100 | ResNext Models | CIFAR-10 | CIFAR-100 |
|---|---|---|---|---|---|
| He et al. (2016)* | 5.84 | 22.15 | Xie et al. (2017)* | 5.03 | 21.24 |
| DBT (5/5) | **5.25** | **21.53** | DBT (5/5) | **4.68** | **19.79** |
| ResNet (4/5) | 5.89 | 24.23 | ResNext (4/5) | 4.93 | 23.52 |
| DBT (4/5) | **5.36** | **23.98** | DBT (4/5) | **4.76** | **22.56** |
| ResNet (3/5) | 6.61 | 27.99 | ResNext (3/5) | 5.38 | 27.25 |
| DBT (3/5) | **5.44** | **26.81** | DBT (3/5) | **4.77** | **26.04** |
| ResNet (2/5) | 7.06 | 33.81 | ResNext (2/5) | 5.85 | 33.62 |
| DBT (2/5) | **5.94** | **31.95** | DBT (2/5) | **5.05** | **30.73** |
| ResNet (1/5) | 8.20 | 47.43 | ResNext (1/5) | 7.24 | 48.05 |
| DBT (1/5) | **6.86** | **43.65** | DBT (1/5) | **6.05** | **42.56** |

Table 8: Comparison of Top-1 error rates (%) for CIFAR-10 and CIFAR-100 datasets w/o DBT.* denotes our implementation. (x/5) denotes the fraction of training data used for training that model.

| Method | VoxC (top 1) | VoxC (top 5) | Librispeech | VCTK | ELSDSR |
|---|---|---|---|---|---|
| VGG-M CNN | 80.5 | 92.1 | 93.12 | 82.52 | 79.98 |
| VGG-M CNN-DBT | **82.3** | **95.8** | **95.62** | **88.14** | **81.56** |

Table 9: Accuracy of speaker identification (%) for different datasets.

| Method | CIFAR-10 | CIFAR-100 |
|---|---|---|
| ResNet-Back | 5.65 | 21.84 |
| ResNet-DBT | **5.25** | **21.53** |
| ResNext-Back | 4.97 | 21.65 |
| ResNext-DBT | **4.68** | **19.79** |

Table 10: Comparison of top-1 error rates on CIFAR-10 and CIFAR-100 using an additional background class vs DBT.

| Method | LFW | CALFW | CPLFW | CFP-FP | AgeDb-30 |
|---|---|---|---|---|---|
| ResNet+mod | 99.16 | 91.46 | 86.11 | 93.81 | 92.71 |
| ResNet-DBT+mod | **99.65** | **95.05** | **90.08** | **96.20** | **95.87** |

Table 11: Ablation study on the verification performance of adding background class to the model on MS1MV2 dataset.

We randomly selected a fraction of the training data to be our training set, e.g., $k/5$ of dataset means that we only used $k$ fifth of total samples for training. From first row of Table 8, we find that models trained with DBT show **0.59**% and **0.35**% improvement on CIFAR-10, **0.62**% and **1.45**% improvement on CIFAR-100 over baseline models for ResNet-110 and ResNext-101 architectures, respectively. Furthermore, using partial training data with our DBT method achieves superior results (as high as 5.49 % on ResNext (1/5) CIFAR-100) as compared to normal training. Table 6 shows the results on Imagenet. We see that DBT improves the accuracy by **0.28**% on Top-1 accuracy. This shows that the DBT method consistently improves the results on both small and large datasets.

SPEAKER IDENTIFICATION

We consider the problem of speaker identification using the VGG-M (Chatfield et al. (2014)) model. We set PEF as the first two CNN layers and DDF as the remaining CNN layers. $\mathcal{L}_{ABC}$ and $\mathcal{L}_{MC}$ are defined to be the cross-entropy loss. The ABC-noise is generated from the silence intervals of VoxCeleb (Nagrani et al. (2017)) augmented with Gaussian noise with variance one. The input to the model is the short-time Fourier transformation of speech signals with a hamming sliding window of width 25 ms and step 10 ms. Table 9 provides the accuracies of VGG-M model trained with and without DBT on VoxCeleb, Librispeech (Panayotov et al. (2015)), VCTK (Veaux et al. (2016)) and ELSDR (L. (2004)) datasets. Table 9 shows that the trained models using DBT significantly improves the accuracy (as high as **5.62%**) for all datasets. Implementation details are provided in the appendix.

MISCELLANEOUS EXPERIMENTS

In this section, we experiment with the naive way of using background data by considering non-faces as a separate class in the final classification layer. For face recognition, Table 11 shows the results of training with an additional background class on MS1MV2 dataset with and without using DBT. ResNet+mod refers to a model trained with ArcFace loss and $n + 1$ classes where the additional class corresponds to the non-faces. ResNet-DBT+mod refers to a model trained with both DBT and the additional non-face class. We find that adding the additional non-face class hurts the performance of the model whereas ResNet-DBT+mod improves the results significantly relative to ResNet+mod model. Since the non-face dataset is sampled from a wide range of a family of distributions compared with faces, it has a larger range of unknown parameters, then the sufficient statistic of them should be larger than the sufficient statistics of face data. Thus, when we restrict faces and non-faces on the surface of a hypersphere, the non-face data is more spread on the surface compared with each of the other face classes. We demonstrate this effect with the help of a toy example in Fig. 6 in the appendix. We also conduct this experiment on CIFAR-10/CIFAR-100 and report it in Table 10. We see that naively incorporating the background class is inferior to DBT showing that DBT is an effective technique to utilize background data to boost the performance of classification models.

## 6 CONCLUSION

In this paper, we presented a detailed theoretical analysis of the dual relationship between estimating the unknown pdf parameters and classification accuracy. Based on the theoretical study, we presented a new method called DBT using ABC-noise data for improving in-distribution classification accuracy. We showed that using ABC-noise data helps in better estimation of unknown parameters of pdf of input data and thereby improves the feature representations and consequently the accuracy in image classification, speaker classification, and face recognition benchmarks. It also augments the training data when only limited labeled data is available by improving accuracy. We showed that the concept of DBT is generic and generalizes well across domains through extensive experiments using different model architectures and datasets. Our framework is complementary to existing training methods and hence, it can be easily integrated with current and possibly future classification methods to enhance accuracy. In summary, the proposed DBT method is a powerful technique that can augment limited training data and improve classification accuracy in deep neural networks.

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

# APPENDIX

## IN-DOMAIN FAMILY OF PDFS AND THE EXTENDED FAMILY OF DISTRIBUTIONS

In this section, we discuss about background/noise and in-domain data points and their corresponding distributions to clarify the definition of those concepts in this paper. Consider a random vector denoted by $\mathbf{s}$. Assume that the corresponding distribution is Gaussian with mean and variance given by $\alpha \neq 0$ and $\sigma = 1$, respectively. Now, assume that we observed $\mathbf{x} = \mathbf{s} + \mathbf{n}$, where the pdf of $\mathbf{n}$ is assumed to be Guassian with zero mean and variance $\sigma_n^2$, hence the pdf of $\mathbf{x}$ is Gaussian with mean $\alpha$ and variance $1 + \sigma_n^2$. Here, $\mathbf{n}$ is the background or noise data and the vector of unknowns is given by, $\boldsymbol{\theta} = [\alpha, \sigma_n^2]$. The *in-domain* family of pdfs for $\mathbf{x}$ is then given by $\mathcal{P}_{\mathbf{x}} = \{\mathcal{N}(\alpha, 1 + \sigma_n^2) | \alpha \neq 0, \sigma_n^2 > 0\}$. If we include the family of pdf of $\mathbf{n}$ to $\mathcal{P}_{\mathbf{x}}$, then we can extend $\mathcal{P}_{\mathbf{x}}$ as $\mathcal{P} = \{\mathcal{N}(\alpha, 1 + \sigma_n^2) | \alpha \in \mathbb{R}, \sigma_n^2 > 0\}$. So $\mathcal{P}$ is the union of family of pdfs of in-domain data points and noise/background data. From estimation theory, we know that the sufficient statistics and the unknown parameters of $\mathcal{P}$ can also represent the sufficient statistics and the unknown parameters of $\mathcal{P}_{\mathbf{x}}$. In other words, an estimation of $\alpha$ can help us detect if the observed data point is from $\mathbf{s} + \mathbf{n}$ or $\mathbf{n}$ by comparing it with a threshold. Thus, estimating the unknown parameters of the family of pdfs using $\mathcal{P}$ can provide more information about the observed data useful for tasks such as classification.

In general, we can assume that a generalized family of pdfs is given by the family of pdf of noise or background along with the family of pdfs of in-domain data. Hence, estimating from the extended

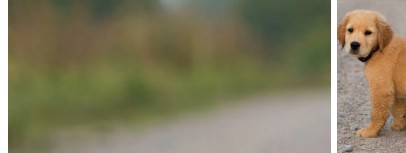

Figure 5: In-domain data point versus background data point. The background is cropped from the in-domain image and provides complementary information to the main data, thereby we can provide a better estimation of the pdf parameters of in-domain data.

family of distribution can provide more information about the in-domain distribution. Let us consider that the pdf of in-domain data points is given by $p_{\mathbf{x}}(\mathbf{x}, [\boldsymbol{\theta}_s, \boldsymbol{\theta}_n])$ and the pdf of noise/background is given by $p_{\mathbf{n}}(\mathbf{x}, \boldsymbol{\theta}_n)$, so the extended pdf can be represented by

$$h(p_{\mathbf{n}}(\mathbf{x}, \boldsymbol{\theta}_n), p_{\mathbf{x}}(\mathbf{x}, [\boldsymbol{\theta}_s, \boldsymbol{\theta}_n])),$$

where $h$ is a function that combines two pdfs in a general structure. So a general family of distribution can be denoted as follows:

$$\mathcal{P} = \{h(p_{\mathbf{n}}(\mathbf{x}, \boldsymbol{\theta}_n), p_{\mathbf{x}}(\mathbf{x}, [\boldsymbol{\theta}_s, \boldsymbol{\theta}_n])) | \boldsymbol{\theta} := [\boldsymbol{\theta}_s, \boldsymbol{\theta}_n] \in \Theta_{s,n}\},$$

where $\boldsymbol{\theta}$ is defined as a new set of parameters in a higher dimension and $\Theta_{s,n}$ are set of all possible $[\boldsymbol{\theta}_s, \boldsymbol{\theta}_n]$ that belongs to $p_{\mathbf{n}}$ and $p_{\mathbf{x}}$. The extended family of pdf provides more information about the nuisance parameters of pdf of in-domain datapoints. Inspired by this observation, we develop our detection booster training method using background/noise data. Figure 5 shows an example of background and in-domain data point.

PROOF OF THEOREM 1

Let $t_i(\cdot)$ denote deterministic discriminative function of $\Theta_i$. Since the efficient minimum variance and unbiased estimation of $t_i(\boldsymbol{\theta})$ exists, we have

$$\frac{\partial \ln(p(\mathbf{x}, \boldsymbol{\theta}))}{\partial t_i(\boldsymbol{\theta})} = I_{t_i}(\boldsymbol{\theta})(\widehat{t_i}(\mathbf{x}) - t_i(\boldsymbol{\theta})), \tag{4}$$

where $\widehat{t_i}(\mathbf{x})$ is the minimum variance and unbiased estimation of $t_i(\boldsymbol{\theta})$ using the data point $\mathbf{x}$ and $I_{t_i}(\mathbf{x})$ is the Fisher information function of $t_i(\boldsymbol{\theta})$, which is given by

$$I_{t_i}(\boldsymbol{\theta}) = \frac{\partial t_i(\boldsymbol{\theta})}{\partial \boldsymbol{\theta}}^T \mathbf{I}(\boldsymbol{\theta}) \frac{\partial t_i(\boldsymbol{\theta})}{\partial \boldsymbol{\theta}} \geq 0,$$

where $^T$ denotes the transpose and $\mathbf{I}(\boldsymbol{\theta})$ is the Fisher information matrix of $\boldsymbol{\theta}$. Now we show that the log-likelihood ratio is an increasing function in $\widehat{t_i}(\mathbf{x})$. Note that $I_{t_i}(\boldsymbol{\theta}) \geq 0$ (Lehmann & Casella (2006)).

On the other hand, we have $\mathrm{d}\ln(p(\mathbf{x}, \boldsymbol{\theta})) = \sum_j \frac{\partial \ln(p(\mathbf{x}, \boldsymbol{\theta}))}{\partial \theta_j} \mathrm{d}\theta_j$, therefore,

$$\ln(p(\mathbf{x}, \boldsymbol{\theta})) + k(\mathbf{x}) = \sum_j \int \frac{\partial \ln(p(\mathbf{x}, \boldsymbol{\theta}))}{\partial \theta_j} \mathrm{d}\theta_j = \sum_j \int \frac{\partial \ln(p(\mathbf{x}, \boldsymbol{\theta}))}{\partial t_i(\boldsymbol{\theta})} \frac{\partial t_i(\boldsymbol{\theta})}{\partial \theta_j} \mathrm{d}\theta_j =$$

$$\int \frac{\partial \ln(p(\mathbf{x}, \boldsymbol{\theta}))}{\partial t_i(\boldsymbol{\theta})} \sum_j \frac{\partial t_i(\boldsymbol{\theta})}{\partial \theta_j} \mathrm{d}\theta_j = \int \left(I_{t_i}(\boldsymbol{\theta})(\widehat{t_i}(\mathbf{x}) - t_i(\boldsymbol{\theta}))\right) \sum_j \frac{\partial t_i(\boldsymbol{\theta})}{\partial \theta_j} \mathrm{d}\theta_j = \alpha(\boldsymbol{\theta})\widehat{t_i}(\mathbf{x}) - \beta(\boldsymbol{\theta}) \tag{5}$$

where the third equality is archived based on the third property of $t_i(\cdot)$ in its definition and the forth equality is given by replacing (4; $k(\mathbf{x})$ is the constant of integration. Finally, the last equality is given by defining the following terms

$$\alpha(\boldsymbol{\theta}) := \int I_{t_i}(\boldsymbol{\theta}) \sum_j \frac{\partial t_i(\boldsymbol{\theta})}{\partial \theta_j} \mathrm{d}\theta_j, \qquad \beta(\boldsymbol{\theta}) := \int I_{t_i}(\boldsymbol{\theta}) t_i(\boldsymbol{\theta}) \sum_j \frac{\partial t_i(\boldsymbol{\theta})}{\partial \theta_j} \mathrm{d}\theta_j, \tag{6}$$

thus $\frac{\mathrm{d}\alpha(\boldsymbol{\theta})}{\mathrm{d}t_i(\boldsymbol{\theta})} = I_{t_i}(\boldsymbol{\theta}) \geq 0$, i.e., $\alpha(\boldsymbol{\theta})$ is increasing in $t_i(\boldsymbol{\theta})$. Since, $t_i$ is a deterministic discriminative function of $\Theta_i$, so for each $j \neq i$ and $\boldsymbol{\theta_i} \in \Theta_i$ and $\boldsymbol{\theta_j} \in \Theta_j$, we have $t_i(\boldsymbol{\theta_i}) > t_i(\boldsymbol{\theta_j})$, therefore

513    $\alpha(\boldsymbol{\theta}_i) \geq \alpha(\boldsymbol{\theta}_j)$. The later inequality is achieved based on the increasing property of $\alpha(\boldsymbol{\theta})$ with
514    respect to $t_i(\boldsymbol{\theta})$.

515    Using (5), the log likelihood ratio of class $i$ against the rest of classes is given by LLR :=
516    $\ln(p(\mathbf{x}, \boldsymbol{\theta}_i)) - \ln(p(\mathbf{x}, \boldsymbol{\theta}_j))$, so we have LLR $= \big(\alpha(\boldsymbol{\theta}_i) - \alpha(\boldsymbol{\theta}_i)\big)\widehat{t}_i(\mathbf{x}) - \big(\beta(\boldsymbol{\theta}_i) - \beta(\boldsymbol{\theta}_j)\big)$. LLR
517    depends on $\mathbf{x}$ only via $\widehat{t}_i(\mathbf{x})$ and since for each $j \neq i$ and $\boldsymbol{\theta_i} \in \Theta_i$ and $\boldsymbol{\theta_j} \notin \Theta_i$, $\alpha(\boldsymbol{\theta}_i) - \alpha(\boldsymbol{\theta}_i) > 0$,
518    then LLR is increasing in $\widehat{t}_i(\mathbf{x})$. $\square$

519    PROOF OF THEOREM 2

The probability of true positive of class $i$ of $r_j$ is given by

$$P_{tp,i,j} = \mathrm{Pr}_{\boldsymbol{\theta}}(h_j(\mathbf{x}) > \tau) = 1 - F_{j_\theta}(\tau),$$

where $F_{i_{\boldsymbol{\theta}}}(\cdot)$ denotes the Cumulative distribution function (CDF) of $h_j$. Since the probability of true positive of class $i$ of $r_1$ is greater than $r_2$ for all $\tau$, $F_{1_\theta}(\tau) < F_{2_\theta}(\tau)$, for all $\tau$. Now we define a function as follows

$$u(\tau, \boldsymbol{\theta}) := F_{2_\theta}(\tau) - F_{1_\theta}(\tau).$$

Since the CDFs are increasing in $\tau$ and tend to 1 and the number of inflection points of these CDFs are bounded, there is an $h_{\min}$ such that, for $\tau > h_{\min}$, such that $u(\tau, \theta)$ is a monotonically decreasing function in $\tau$. Thus for any $\boldsymbol{\theta}$ that satisfies $d(\boldsymbol{\theta}) > h_{\min}$ we have

$$u(d(\boldsymbol{\theta}) + \epsilon, \theta) < u(d(\boldsymbol{\theta}) - \epsilon, \theta).$$

520    Replacing $u(h, \theta) = F_{2_\theta}(h) - F_{1_\theta}(h)$ in the last inequality, we have

$$F_{2_\theta}(d(\boldsymbol{\theta}) + \epsilon) - F_{1_\theta}(d(\boldsymbol{\theta}) + \epsilon) < F_{2_\theta}(d(\boldsymbol{\theta}) - \epsilon) - F_{1_\theta}(d(\boldsymbol{\theta}) - \epsilon) \Rightarrow \qquad (7)$$

$$F_{2_\theta}(d(\boldsymbol{\theta}) + \epsilon) - F_{2_\theta}(d(\boldsymbol{\theta}) - \epsilon) < F_{1_\theta}(d(\boldsymbol{\theta}) + \epsilon) - F_{1_\theta}(d(\boldsymbol{\theta}) - \epsilon). \qquad (8)$$

521    Based on the definition of CDF, we have

$$\mathrm{Pr}_{\boldsymbol{\theta}}\Big(|h_2(\mathbf{x}) - d(\boldsymbol{\theta})| < \epsilon\Big) = \mathrm{Pr}_{\boldsymbol{\theta}}\Big(d(\boldsymbol{\theta}) - \epsilon < h_2(\mathbf{x}) < d(\boldsymbol{\theta}) + \epsilon\Big) <$$

$$\mathrm{Pr}_{\boldsymbol{\theta}}\Big(d(\boldsymbol{\theta}) - \epsilon < h_1(\mathbf{x})) < d(\boldsymbol{\theta}) + \epsilon\Big) = \mathrm{Pr}_{\boldsymbol{\theta}}\Big(|h_1(\mathbf{x}) - d(\boldsymbol{\theta})| < \epsilon\Big). \qquad (9)$$

522    $\square$

523    PROOF OF THEOREM 3

524    First, we prove the following claim,
525    Claim: For any open set, there exists a set of disjoint countable open balls such that their union equals
526    the origin open set.
527    Proof of claim: Consider an open set $\mathcal{O}$, and also consider $x_0 \in \mathcal{O}$, such that $B(x_0, r_0) \subseteq \mathcal{O}$
528    and $r_0$ is the greatest possible radius between all possible open balls in $\mathcal{O}$, where $B(x_0, r_0)$ is the
529    open ball with radius $r_0$ at point $x_0$. Now, we define $x_1 \in \mathcal{O} - \overline{B(x_0, r_0)}$, where $\overline{B(x_0, r_0)}$ is
530    the closure of $B(x_0, r_0)$, as the point with greatest radius in $\mathcal{O} - \overline{B(x_0, r_0)}$ and similarly $x_i \in$
531    $\mathcal{O} - \cup_{k=0}^{i-1}\overline{B(x_k, r_k)}$ such that $B(x_i, r_i)$ provides the greatest radius in $\mathcal{O} - \cup_{k=0}^{i-1}\overline{B(x_k, r_k)}$. So
532    we have $\mathcal{O} = \cup_{k=0}^{\infty}B(x_k, r_k)$. This is because, if the latest equality is not valid, then there exists
533    an open ball in $\mathcal{O} - \cup_{k=0}^{\infty}\overline{B(x_k, r_k)}$ hence another open ball with greatest radius will be added to
534    $\cup_{k=0}^{\infty}B(x_k, r_k)$, which has a contradiction with the definition of $\cup_{k=0}^{\infty}B(x_k, r_k)$. The claim is proven
535    at this point.

Now, we show the true positive probability of $r_1$ is greater than $r_2$. Let $\Theta_m'$ be the set of interior points of $\Theta_m$, then, there exists a union of disjoint open balls such that $\Theta_m' = \cup_{k=0}^{\infty}B(x_k, r_k)$. From assumptions in the theorem, we have $\mathrm{Pr}(\|\widehat{\boldsymbol{\theta}}_1 - \boldsymbol{\theta}\| \leq \epsilon) \geq \mathrm{Pr}(\|\widehat{\boldsymbol{\theta}}_2 - \boldsymbol{\theta}\| \leq \epsilon)$, then

$$\mathrm{Pr}_{\boldsymbol{\theta}}(\widehat{\boldsymbol{\theta}}_1 \in B(x_k, r_k)) \geq \mathrm{Pr}_{\boldsymbol{\theta}}(\widehat{\boldsymbol{\theta}}_2 \in B(x_k, r_k)),$$

536    where $\boldsymbol{\theta} \in \Theta_m$. Based on the claim we have

$$\mathrm{Pr}_{\boldsymbol{\theta}}(\widehat{\boldsymbol{\theta}}_1 \in \Theta_m') \geq \mathrm{Pr}_{\boldsymbol{\theta}}(\widehat{\boldsymbol{\theta}}_2 \in \Theta_m'). \qquad (10)$$

Moreover, based on definition of $r_i$, the true positive probability of class $m$ is given by

$$p_{tp,i} = \mathrm{Pr}_{\boldsymbol{\theta}}(\widehat{\boldsymbol{\theta}}_i \in \Theta_m) = \mathrm{Pr}_{\boldsymbol{\theta}}(\widehat{\boldsymbol{\theta}}_i \in \Theta_m') + \mathrm{Pr}_{\boldsymbol{\theta}}(\widehat{\boldsymbol{\theta}}_i \in \Theta_m - \Theta_m'),$$

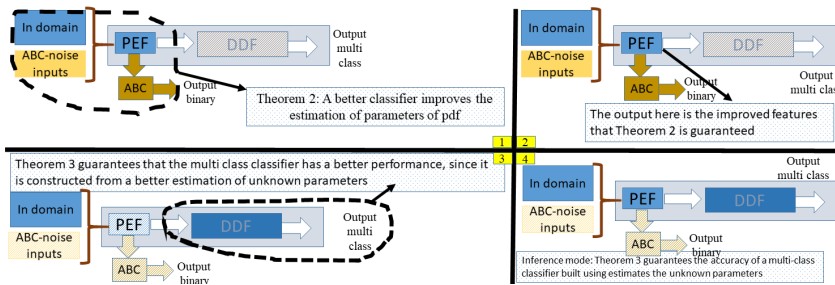

Figure 6: Relationship between the theorems in Section 3 and the proposed method in Section 4.

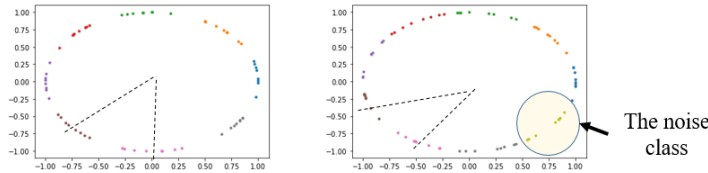

The noise class

Figure 7: Feature distance between different classes with and without additional background class for a toy example. Left: Contains 8 classes and the feature separation is visibly larger; Right: Contains an additional noise class that decreases the feature distance for all the other classes.

for $i = 1, 2$. Additionally, from the Cauchy–Schwarz inequality, we have

$$\text{Pr}_{\boldsymbol{\theta}}(\widehat{\boldsymbol{\theta}}_i \in \Theta_m - \Theta'_m) \leq \mu_l(\Theta_m - \Theta'_m) = 0,$$

So, $p_{tp,i} = \text{Pr}_{\boldsymbol{\theta}}(\widehat{\boldsymbol{\theta}}_i \in \Theta'_m)$ and from (10) the true positive probability of class $i$ of $r_1$ is greater than $r_1$.

The error probability of $r_j$ is given by $p_{er,j} = 1 - \sum_{i=1}^{n} P_i P_{tp,i,j}$, where $P_i$ is the prior probability of class $i$. Therefor, $p_{er,1} \leq p_{er,2}$. $\square$

CONNECTING THE THEOREMS WITH THE PROPOSED METHOD

Fig. 6 shows the connection between the proposed theorems and the approach. In part 1, Theorem 2 connects the estimation of unknown parameters to the auxiliary classifier. In part 2, the learned features are passed to a decision making network (result of Theorem 2). In part 3, Theorem 3 guarantees that the multi-class classifier outperforms other classifiers, because it is using the features from a better estimation of unknown parameters of pdf.

TOY EXAMPLE:

We demonstrate the effect of adding background class to the original classifier with a toy example and visualize it in Fig. 7. In this example, the input is a sequence of binary bits ($+1$ and $-1$) with length 3 in white Gaussian noise. the classifier is constructed using two fully connected layers with sigmoid and the last layer is normalized on unit circle. As seen from Fig. 7, adding an additional noise class visibly reduces the feature separation between all the other classes.

IMPLEMENTATION DETAILS

FACE RECOGNITION

We use Tensorflow (Abadi et al. (2015)) to conduct all our experiments. We train with a batch size of 256 on two NVIDIA TeslaV100 (32G) GPUs. We train our models following small (less than 1M training images) and large (more than 1M training images) protocol conventions. We use CASIA-Webface (Yi et al. (2014)) dataset for small protocol and MS1MV2 dataset for the large protocol. We use ResNet-50 (He et al. (2016)) and ResNet-100 models for small and large protocols, respectively. The PEF is selected as the first three layers. Following (Deng et al. (2019)), we apply

BN (Ioffe & Szegedy (2015)), dropout (Srivastava et al. (2014)) to the last feature map layer followed by a fully connected layer and batch normalization to obtain the 512-D embedding vector. We set the feature scale $s$ parameter to 64 following (Wang et al. (2018); Deng et al. (2019)) and set the margin parameters $(m_1, m_2, m_3)$ to (1, 0.5, 0), respectively. For small scale protocol, we start the learning rate at 0.01 and divide the learning rate by 10 at 40K, 80K, and 100K iterations. We train for 120K iterations. For large scale protocol, we start the learning rate at 0.01 and divide the learning rate by 10 at 80K, 100K, and 200K iterations. We train for 240K iterations. We use Momentum optimizer and set the momentum to 0.9 and weight decay to 5e-4. We use the feature centre of all images from a template or all frames from a video in order to report the results on IJB-B, IJB-C and YTF datasets. For ABC-noise data, we cropped background images patches from MS1MV2 (Guo et al. (2016)) dataset and cropped hard examples from the Caltech-101 (F. F. Li et al. (2004)) dataset plus a few open sourced images (animal faces) using MTCNN (Zhang et al. (2016a)) face detector. We generated roughly 500K non-face images for training the ABCs.

### SPEAKER IDENTIFICATION

L2 loss and dropout with a rate of 0.2 are applied during training for generalization. The ABC-noise is collected form silence intervals of the VoxCeleb dataset, where an energy-based voice activity detection (VAD) is applied to detect the silence intervals. To augment the ABC-noise, Gaussian noise is added to the silence intervals. Each batch size is set to 64 and the optimizer is ADAM with a learning rate of 0.001. The VoxCeleb dataset is trained for 11 epochs and the other datasets are trained for 6 epochs.

### LFW AND YTF DATASETS

LFW database contains the annotations for 5171 faces in a set of 2845 images taken from the Faces in the Wild data set (Berg et al. (2004)). YouTubeFaces (Wolf et al. (2011)) contains 3,425 videos of 1,595 people. Following the standard convention, we report the results on 5000 video pairs using unrestricted with labeled outside data protocol.

### IJB-B AND IJB-C DATASETS

The IJB-B contains 1,845 subjects with 21.8K still images and 55K frames from 7,011 videos. In total, there are 12,115 templates with 10,270 genuine matches and 8M impostor matches. The IJB-C dataset (Maze et al. (2018)) is a further extension of IJB-B, having 3,531 subjects with 31.3K still images and 117.5K frames from 11,779 videos. In total, there are 23, 124 templates with 19,557 genuine matches and 15, 639K impostor matches.

