# OpenReview forum: "Detection Booster Training: A detection booster training method for improving the accuracy of classifiers."
_ICLR.cc/2021/Conference — Reject_

### Official Review · AnonReviewer1 · 2020-10-25
**Review by AnonReviewer1**

**Rating:** 4
**Confidence:** 5

**Review:**

This submission proposes a training strategy that leverages background/noise data to learn robust representations. Experimental results show that the use of this strategy generally leads to improved performance.

Quality: The quality of this submission is lacking in several respects.

1) You abstract states that there has been not much work done on using background/noise data whilst estimating model parameters. This is not true. What about NCE and all the follow up work?
2) What you are describing is a feature extractor that is using one objective and classifier that uses another objective. There is prior work in this area. However, your related work section suggests something entirely different.
3) You are cramming way too much into 8 pages.
4) Section 3, 4 and 5 do not work together. Section 3 talks about Cramer-Rao bounds and Lebesgue measures while section 4 shows something that looks like a multi-task training objective. Section 5 does not have baselines other than the trivial one of not using your approach.

Clarity: The clarity in this submission is lacking. Given Section 4, the point of sections 2 and 3 is very unclear.

Originality: It is hard to judge originality of this submission given the way it is presented. Which theorems are your contribution? Which of many statements you make in sections 2, 3 and 4 are your contributions?

Significance: It is hard to judge the significance of this submission given the way it is presented.

Pros: This submission contains evaluation across many modalities.

Cons: One-sided presentation with links to existing works missing.

---

> ### Author Response · Authors · 2020-11-17
> **Response to Reviewer 1**
>
> We thank the reviewer for their detailed and constructive comments.
>
> "Your abstract states that there has been not much work done on using background/noise data whilst estimating model parameters. This is not true. What about NCE and all the follow up work?"
>
> Thanks for pointing it out. We have added the NCE and other works into the related works section of the paper and compared it with our method. Further, some other methods such as Generalized Likelihood Ratio (GLR) is also added to the related works. Our approach is novel in terms of making a connection between the estimation of parameters of pdf of data and use them in building a deep neural network for classification. Moreover, we showed that estimating parameters of pdfs provide a sufficient statistic for the classification problem. Hence, in our work instead of directly estimating the pdf of data or noise, we improve the feature layers that are built using the estimation of unknown parameters of pdfs of input data.
>
> "What you are describing is a feature extractor that is using one objective and classifier that uses another objective. There is prior work in this area. However, your related work section suggests something entirely different."
>
> As mentioned above, we included the missing related works. We proposed a method using detecting in-domain data from noise/background and we showed that having that detector in early layers can improve the overall accuracy of classification.
>
> "You are cramming way too much into 8 pages."
>
> We wanted to show that our proposed approach works well for different vision and speech tasks.
>
> "Section 3, 4 and 5 do not work together. Section 3 talks about Cramer-Rao bounds and Lebesgue measures while section 4 shows something that looks like a multi-task training objective."
>
> With all due respect, Section 3 establishes the relationship between the estimation of parameters of pdf and the accuracy of classifiers, it does not talk about Lebesgue measures. All related real analysis concepts such as Lebesgue measures and Borel sets are used in this paper to construct our proposed theorems in a proper structure with sufficient conditions in place. Moreover, the Cramer Rao bound plays an important role in proposing Theorems 2 and 3 of the paper which are the foundation of this paper and the proposed method. We have added Figure 6 in the Appendix to depict the relationship between our proposed method and the theorems visually. Please refer to Section "CONNECTING THE THEOREMS WITH THE PROPOSED  METHOD" in the Appendix.
>
>
> "Section 5 does not have baselines other than the trivial one of not using your approach."
>
> To the best of our knowledge, we are the first to propose this method and so for fair comparisons, we only include the baselines without DBT.
>
> "It is hard to judge originality of this submission given the way it is presented. Which theorems are your contribution? Which of many statements you make in sections 2, 3 and 4 are your contributions?"
>
> The theorems in Section 3  are all our contributions. We prove the theorems in the appendix. Based on the theorem results, we propose the auxiliary binary classifier method in Section 4 to improve classification accuracies.
>
> "One-sided presentation with links to existing works missing."
>
> We have included the related works in the revised version and hopefully addressed the other concerns.

---

### Official Review · AnonReviewer2 · 2020-10-28
**improving robustness of DNN classification w/ auxiliary classifiers for distractors**

**Rating:** 6
**Confidence:** 3

**Review:**

To improve multi-class classification problem using DNNs, authors propose to do multi-task learning of solving another auxiliary tasks which tells if data points are just noise/distractors or not. It is common to have some data which is non-representative of good training samples but they can be used (under DBT) to make first few layers (and thus all) of network more robust.
Results are satisfactory but:
1. Can this idea not be done simply by having one more classification unit which detects if x is in-domain or noise? I believe that experiment should be done. That way, whole network can be trained for this. I believe my suggestion is different from "DBT(ABC-last)" of appendix.
2. The multi-task formulation should have a hyper-parameter lambda in Equation 2. Is there a reason it is set to 1? I think it is standard to experiment with that. Perhaps equal weightage to two tasks is not optimal.
3. It is not clear if I use a regularization technique like MixUp or label smoothing, will I still see improvement?

I am not much aware of related work BTW.

---

> ### Author Response · Authors · 2020-11-17
> **Response to Reviewer 2**
>
> We thank the reviewer for their detailed and constructive comments.
>
> "Can this idea not be done simply by having one more classification unit which detects if x is in-domain or noise? I believe that experiment should be done. That way, whole network can be trained for this."
>
> We have added this experiment in the revision. Please refer to our answer to Reviewer 5 on a similar question. We have shown that adding an extra class decreases the accuracy of classifiers since the extra class can shrink the last layer manifold. Please see Tables 10 and 11 and Figure 6 in the Appendix in the revised version.
>
> "The multi-task formulation should have a hyper-parameter lambda in Equation 2. Is there a reason it is set to 1?"
>
> We empirically set lambda=1 to balance the two loss terms. We updated the equation in the revision. The results were not better with other hyperparameter settings in our experiments.
>
> "It is not clear if I use a regularization technique like MixUp or label smoothing, will I still see improvement?"
>
> Mixup or label smoothing is complementary to our proposed method. We have not experimented with it but its a good idea to try in the future.

---

### Official Review · AnonReviewer5 · 2020-11-06
**experiments are not sufficient**

**Rating:** 4
**Confidence:** 4

**Review:**

This paper proposes a training method for classification, with the goal of training with less data. The proposal is to train an auxiliary classifier at the same time. The auxiliary classifier and the main classifier share the early layers. The auxiliary classifier is a binary classifier that discriminates training data versus background/noise data. The proposed method is evaluated on image and speech classification tasks.

On the positive side, the experimental results show consistent benefit across tasks. The facial recognition comparisons should be taken with a grain of salt, because of how the background data were collected (see page 7 and appendix). Other than that, the results seem quite consistent.

However, I have substantial concerns. The experiments are not sufficient to support the claims: there is little results on training with less data; certain critical experiments are missing; some comparisons are not fair. The theoretical content is detached from the proposed method and hence not important. I'll elaborate in the following.
There are three angles to look at the proposed method:

1. Data augmentation. Part of the proposed method is building an extra class of training data that are background or noise. One example is given in Figure 5. It seems feasible for image and speech tasks.

   A critical point, that is missing from this paper, is how would this data augmentation work by itself without the auxiliary classifier. In other words, what if we just train a classifier with n+1 labels, where the extra label's training data are the background or noise data points? At inference time, one simply uses the first n logits. How much of the reported benefits would this simplified method achieve? This would show how important is the data augmentation versus how important is the auxiliary classifier.

   The appendix stated that the background/noise data include animal faces collected from open source images. Therefore, the discussion around Figure 4 and Table 2, which claims robustness advantage over ArcFace, is invalid. It's not a fair comparison when the training data is expanded with a particular emphasis.

2. Semi-supervised learning with less labeled data. It seems that the most natural and meaningful task is where the auxiliary classifier is trained on all in-domain data, both labeled and unlabeled, while the main classifier is trained on a small number of labeled data. It's important to compare with existing methods in this setup.

   Unfortunately this paper does not include such experiments. Table 8 includes reduced-training-set experiments on CIFAR, however 3/5 of the training set is still way too big, and the remaining 2/5 are not utilized as unlabeled in-domain data.

3. Regularization of the early layers. This is the stated rationale for the proposed method. The auxiliary classifier is designed to encourage the early layers to learn more meaningful features, and Section 3 supports this relation.

   Section 3 is very difficult to read. The conclusion is that the cross entropy loss of the auxiliary classifier is positively correlated with the quality of the estimation of unknown parameters of the data distribution, and therefore training with the joint loss of equation (2) is implicitly related to a good estimation of parameters, which in turn is related to better feature representations in the early layers (an implicit assumption here is that good features for the auxiliary binary classification are also good features for the multi-class classification, which could be a question by itself).

   This relation, however, is qualitative rather than quantitative. In other words, neither training nor inference of the proposed method utilizes the content of Section 3. Section 3 mostly serves as a motivational argument. The auxiliary classifier idea is intuitive by itself, and it seems that Section 3 can be removed without affecting the core content of this paper.

   The stated goal of regularization is to learn with less data. However the only results on this subject seems in Table 8 where 3/5 of CIFAR data are used. Table 8 is not the best setup, as mentioned earlier in the second point. Even not considering semi-supervised learning, 3/5 of the CIFAR training data are too many for a training-with-less-data experiment.

Minor points:
-- Figure 3 is not a fair comparison. With the joint training, the roles of ResNet blocks are likely shuffled. Some information that gets processed in later layers during normal training may get moved to early layers in joint training, and vice versa. Hence comparing at a fixed layer is not meaningful.

---

> ### Author Response · Authors · 2020-11-17
> **Response to Reviewer 5**
>
> We thank the reviewer for their detailed and constructive comments.
>
> "This paper proposes a training method for classification, with the goal of training with less data. "
>
> The target of this paper is not training using less data. The goal of the proposed approach is to improve classification accuracies. We observe that training with less data is an auxiliary result that is more beneficial with our method as compared to traditional training recipes. We looked at training deep neural networks from the perspective of parametric estimation and showed that it indeed helps in improving the classification accuracies. Our approach also opens up a new direction of research for improving classification results such as using auxiliary generative methods that we mentioned in the paper. We kept this for future work.
>
> "The experiments are not sufficient to support the claims: there is little results on training with less data; certain critical experiments are missing"
>
> As explained above, the major contribution of the paper is in improving classification accuracies. Superior performance with less training data is a beneficial result that we observed. We conducted extensive experiments on three tasks with more than 30 experimental results to support our claims as shown in the paper. We have also added new experimental results in the revised version as requested by the reviewers.
>
> "The facial recognition comparisons should be taken with a grain of salt..."
>
> The background data collected for face recognition is consistent with our theoretical analysis. The background patches cropped from the images come from the same family of distribution of images but with different parameter sets. For the robustness analysis, we used only 200 animal images as compared to million images from MS1MV2 dataset. For fair comparisons, this data is disjoint from the data that we use for the animal distractor identification experiment.
>
> "In other words, what if we just train a classifier with n+1 labels..."
>
> We have conducted experiments using this method and observed a reduction in accuracy but couldn't include it in the paper for lack of space. Please find the results in the revised version Table 10 and 11. We also visualize the feature space when an additional class is added with a toy example. Please see Figure 7 in the Appendix, we find that the feature distance between classes is reduced by adding a new background class.  So, it hurts the overall accuracy of the classification.
>
>
> "It's not a fair comparison when the training data is expanded with a particular emphasis."
>
> As there were no standard datasets for this task, we had to collect the dataset ourselves. To the best of our knowledge, we are the first to bring out the robustness issues of current face recognition models against animal/non-human distractors and show that our proposed approach implicitly overcomes this issue. We will release the code and datasets after the acceptance of the paper for reproducibility.
>
> "Semi-supervised learning with less labeled data."
>
> Thanks for this suggestion. The semi-supervised idea is interesting and we would like to try it in the future. However, in this paper, we focused only on fully supervised methods.
>
> "Table 8 includes reduced-training-set experiments on CIFAR, however, $3/5$ of the training set is still way too big..."
>
> We have updated Table 8 to include 2/5 and 1/5 training experiments. We observe consistent improvements with DBT as compared to normal training recipe. Please refer to the revision.
>
> "an implicit assumption here is that good features for the auxiliary binary classification are also good features for the multi-class classification, which could be a question by itself"
>
> The assumption holds valid because we consider the background data to be in the same distribution family as the main data. Please refer to Figure 6 of the paper (Appendix), here, we show the relationship between our proposed auxiliary classifier and the theorems (relation between the estimation of parameters of pdf and the accuracy of classifiers) visually. We have proven three theorems to explain this complicated entangled relationship between the theory of estimation and classification.
>
> "In other words, neither training nor inference of the proposed method utilizes the content of Section 3."
>
> Please refer to Figure 6 (Appendix) of the paper for the connection between the proposed method and Section 3. The paper uses concepts from parametric estimation theory to propose a new direction of research for improving classification accuracies. In this paper, we used an auxiliary binary classifier to achieve the same. In the future, we plan to work on auxiliary generative models to improve the classification benchmarks. In fact, Section 3 is proof of why the proposed method should improve the accuracy of classifiers and also can provide important theorems in the relationship between estimation theory and classification methods.

---

> > ### Comment · AnonReviewer5 · 2020-11-17
> > **Table 11 may have issues**
> >
> > It is noted in the text that numbers in Table 11 are bad because "the class of non-faces are much larger than faces". Data size imbalance is an easily fixable problem. So it seems that the current numbers in Table 11 are not informative. This also suggests that the first and third rows in Table 10 might also be worse than they should be.

---

> > > ### Author Response · Authors · 2020-11-17
> > > **Not an issue of data imbalance - We used a balanced data for training**
> > >
> > > Sorry, if the reviewer misunderstood the statement. This issue is not data imbalance. For both Tables 10 and 11, we use the same number of examples for the background class as others (eg., 5000/class for CIFAR-10 and 500 /class for CIFAR-100) to conduct our experiments.
> > >
> > > In the pointed part of the text ("the class of non-faces is much larger than faces"), by large we do not mean the unbalanced datasets. We mean that, since the non-face dataset is sampled from a wide range of a family of distributions compared with faces, it has a larger range of unknown parameters, then the sufficient statistic of them should be larger than the sufficient statistics of face data. Hence, when we restrict faces and non-faces on the surface of a hypersphere, the non-face data is more spread on the surface compared with each of the other face classes.
> > >
> > > We also conduct a simulation with a toy example to show this effect and report it in Figure 7.  From Fig. 7 we see that adding a noise class reduces the feature separation between all the other classes. Since the features in the penultimate layer (which is a linear fully connected layer) are linearly separable, then adding a new class can shrink the space for other classes, which is what we observe in Fig. 7.
> > >
> > > We have rephrased the sentence in the revision to avoid any confusion.

---

### Author Response · Authors · 2020-11-17
**Response to all the reviewers**

We updated the revised manuscript with the details requested by the reviewers. We highlighted the new changes in the "blue" text for ease of viewing.

---

### Decision · Program_Chairs · 2021-01-07
**Final Decision**

**Decision:**

Reject

**Comment:**

In this paper the authors propose an approach to improving the accuracy of the classification problem based on deep neural networks by detecting the in-domain data from background/noise.  The strategy is designed in such a way that the detector and the classifier share the bottom layers of the network.  Theoretical proof is given and experiments are conducted on a variety of datasets.  The novelty of the work is to come up with a better estimate the pdf of the data and use it to help the classification based on the deep neural networks.   There are concerns raised by the reviewers regarding the related work, the exposition and the experimental design.  After the rebuttal from the authors, which is meticulous, some of the issues unfortunately still stand.  The paper needs to make a stronger case in order to be accepted, especially, for instance, the theoretical and empirical comparison with the existing techniques sharing the similar idea.